# AttAnchor: Guiding Cross-Modal Token Alignment in VLMs with Attention Anchors

## Abstract

A fundamental reason for the dominance of attention over RNNs and LSTMs in LLMs is its ability to capture long-range dependencies by modeling direct interactions between all tokens, overcoming the sequential limitations of recurrent architectures. Similarly, a key reason why today's vision–language models (VLMs) hallucinate and underperform pure language models is that they rely on direct concatenation of image and text tokens with a modality-blinded positional encoding, which conveniently adopts the pretrained LLM backbone but forces unnecessary long-distance attention between semantically related tokens across modalities. This underscores the urgent need for mechanisms that efficiently enhance token locality and cross-modal alignment. In response, we propose Attention Anchor, a parameter-free framework that efficiently groups semantically similar tokens across modalities, improving cross-modal locality. By inserting text tokens near relevant visual patches, we create semantic signposts that reveal true content-based cross-modal attention scores, guiding the model to focus on the correct image regions for tasks such as VQA, MMBench and POPE. This improves answer accuracy and reduces hallucinations without disrupting the prompt's semantic flow. AttAnchor achieves improvements across 13/15 different metrics and benchmarks, including up to 32% gains on reasoning tasks and up to 15% improvements on hallucination benchmarks. AttAnchor enables TinyLLaVA 1B to outperform much larger models like LLaVA 7B and QwenVL 3B on POPE with only 0.1% inference time overhead. To the best of our knowledge, this work is among the first to investigate mixed-modal token grouping, where text and image tokens are clustered jointly into shared groups rather than being grouped within a single modality or merely aligned post-hoc with additional alignment losses.

## 1 Introduction

Attention mechanisms (Vaswani et al., 2017) have rapidly overtaken RNNs (Rumelhart et al., 1986) and LSTMs (Graves, 2012) as the dominant paradigm for sequence modeling due to their efficiency and ability to capture long-range dependencies. Unlike recurrent models, which process tokens sequentially and suffer from forgetting long-range dependencies and vanishing gradients, attention enables direct interactions between all tokens in parallel, dramatically improving both training speed and representational power. Early research highlighted the severity of this limitation in recurrent models: for instance, Sequence to Sequence Learning with Neural Networks (Sutskever et al., 2014) demonstrated that simply reversing the input sentence significantly improved LSTM translation performance, as this reduced the physical distance between the first input token and the first generated token. This reliance on heuristics to enforce locality underscores a fundamental limitation of recurrent architectures. By contrast, attention enables every token to directly interact with all others in parallel, preserving locality without sacrificing global context, and has therefore become the foundation of modern large-scale language and vision–language models.

Current vision–language models (VLMs) like LLaVA (Liu et al., 2023), LLaVA-NeXT (Liu et al., 2024b), LLaVA-MORE (Liu et al., 2024a), BLIP-2 (Li et al., 2023a), InstructBLIP (Dai et al., 2023), MiniGPT-4 (Zhu et al., 2023), LLaMA-Adapter (Zhang et al., 2023), QwenVL (Wang et al., 2024) and TinyLLaVA (Wang et al., 2023b) and Multimodal LLMs (MLLMs) like NextGPT (Wu et al., 2024), X-LLM (Chen et al., 2023a) and OneLLM (Han et al., 2024) typically adopt a pretrained LLM backbone and simply concatenate tokens from different modalities generated by their

corresponding projectors and encoders (e.g., Vision Transformers (Dosovitskiy et al., 2020)), since breaking the positional prior and retraining the LLM backbone with modality-aware positional encodings (Wang et al., 2025) is prohibitively expensive and might lead to catastrophic forgetting of LLM's pretrained knowledge base (Appendix J). This popular design places semantically related tokens from different modalities far apart in the sequence. Moreover, the direct use of RoPE positional encoding (Su et al., 2024) across modalities distorts content-based attention scores, because relative position is meaningful within a single modality but largely meaningless across modalities—for example, it is uninformative and confusing to claim that an image token is 'further away' from a text token than from an audio token. Such modality-blinded encoding (Press et al., 2021; Chen et al., 2023b; Sun et al., 2022) degrades cross-modal alignment and contributes to hallucination and underperformance compared to pure LLMs.

**Our Approach**: To address the challenge of aligning text and image information in VLMs, we introduce Attention Anchor (AttAnchor), a novel parameter-free architectural framework that reorders tokens by copying and placing text tokens (e.g., "shirt") near semantically similar image patches (e.g., shirt visuals) based on their cosine similarity, as shown in Figure 1. This method acts like adding signposts to a map or placing sticky notes next to relevant pictures in a book: it guides the model's attention to focus on the right image regions when answering questions, such as "is the person skiing or skateboarding?" in Figure 2, without disrupting the original question's meaning. These signposts help reveal the true content-based cross-modal attention scores by reducing positional penalties from standard encodings, enabling deeper transformer layers to leverage the auxiliary information and more accurate text-image interactions for better image understanding. By bringing highly relevant text and image tokens closer together, AttAnchor overcomes the limitations of standard positional encodings, like RoPE, which often weaken connections between distant tokens, leading to confusion or incorrect answers (hallucinations). This enhanced cross-modal alignment strengthens the model's ability to understand images accurately and respond correctly, all while requiring minimal changes to existing VLM architectures and negligible extra computation, making it a practical, plug-and-play solution for improving general image understanding in VLMs and multimodal LLMs, especially under computational constraints.

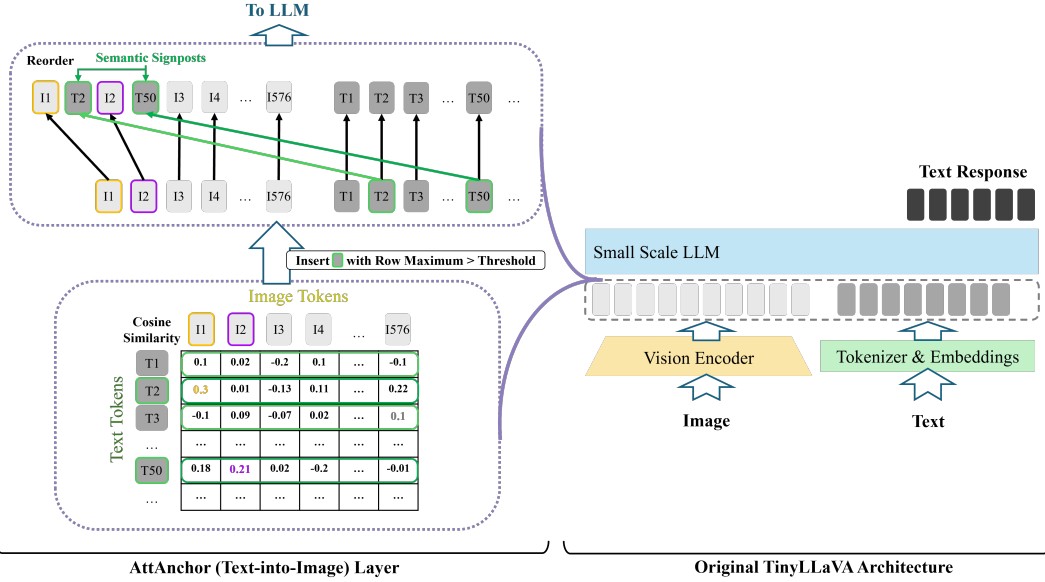

Figure 1: AttAnchor (Text-into-Image): Multimodal Token Reordering in TinyLlava Architecture.

Figure 1 illustrates our approach in a typical vision–language pipeline. Models such as LLaVA or TinyLLaVA concatenate hundreds of image tokens (e.g., 576 from a CLIP vision encoder (Radford et al., 2021)) with a short text prompt (e.g., 50 tokens) into a single flat sequence $(I_1, I_2, \ldots, I_{576}, T_1, T_2, \ldots, T_{50})$. This design forces every transformer layer in the LLM backbone to rely on costly global attention to identify semantic correspondences—for example, aligning

tokens like "person" ($T_{50}$) with person-image patches ($I_2$), but RoPE's positional encoding penalizes distant pairs, dropping their pre-softmax attention score from $1.2$ (relevant) to $-1.7$ (unrelated), introducing significant noise that hinders cross-modal reasoning. Our AttAnchor addresses this by inserting text tokens (e.g., "person") near similar image patches based on cosine similarity, creating semantic signposts (e.g., $T_2$, $T_{50}$) that reveal true content-based attention scores, enabling deeper layers to leverage accurate text-image interactions for robust image understanding while preserving the original text sequence.

Prior work has largely restricted token grouping to the visual modality (Huang et al., 2024; Cao et al., 2023; Fan et al., 2024) or relied on additional fine-grained losses to improve alignment between modalities (Yin et al., 2024; Bica et al., 2024; Mukhoti et al., 2023) . In contrast, we perform mixed-modal token grouping, where text and image tokens are clustered jointly into shared groups as shown in Figure 2 and Appendix E. To the best of our knowledge, this is among the first works to explore explicit cross-modal grouping, enabling more direct multimodal interaction. We perform extensive experiments on TinyLLaVA-1B, LLaVA-7B, and QwenVL-3B across three diverse datasets, evaluating at the best checkpoints. Our study systematically varies LoRA rank, learning rate, batch size, number of epochs, early-stopping patience, and weight decay rate, consistently achieves improvements on 13 of the 15 total evaluated metrics and benchmarks, as shown in Figure 3. These gains span multiple levels—from token-level cross-entropy loss, to sentence-level generation metrics such as BLEU and BERTScore, to reasoning-level benchmarks including VQA, MMBench, and POPE—demonstrating the robustness and generality of our approach. We also propose AttAnchor (Image-into-Text), another novel and reverse strategy that uses image tokens as attention anhors and inserts them into the text sequence, preserving the full image sequence at the start. While this approach enhances cross-modal alignment for larger models like LLaVA-7B, which can leverage their greater capacity to handle disrupted prompt causality, it fragments the text's semantic flow, leading to unstable performance and degraded reasoning capabilities in smaller models like TinyLLaVA-1B.

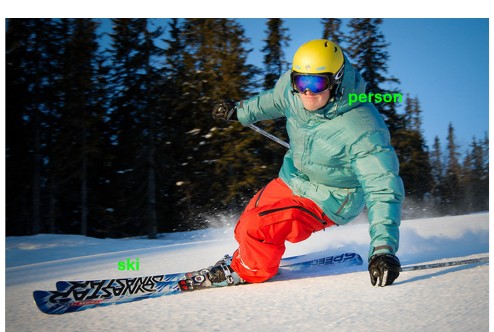

**Prompt:** Is the person in the image skiing or skateboarding?
**Model Answer:** The person in the image is skiing down a snow-covered slope.

Figure 2: Attention Anchors Assisting VLM to Answer Prompts: Key text tokens are inserted near the most relevant image tokens.

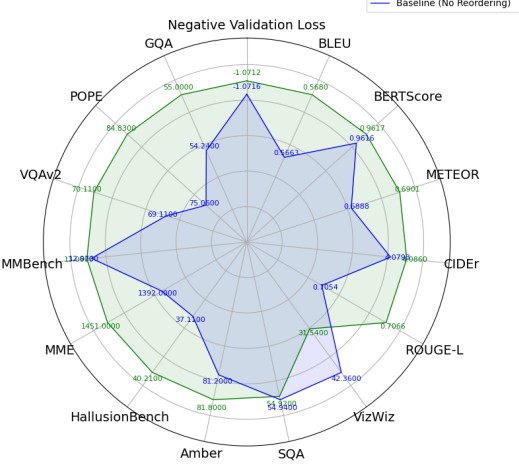

Figure 3: Radar Map Comparison between AttAnchor (Text-into-Image) and Baseline (No Reordering).

Our contributions include:

- **Enhanced Cross-Modal Alignment:** AttAnchor reorders tokens by copying text tokens (e.g., "shirt") near similar image patches, creating semantic signpostthat reveal true content-based attention scores for improved image understanding.
- **Token- and Sentence-Level Improvements:** Reduces cross-entropy loss and improves sentence-level metrics (BLEU, BERTScore, CIDEr, METEOR, ROUGE-L) significantly on captioning and VQA tasks, enhancing text generation quality and semantic alignment.
- **Reasoning-Level Improvements:** Boosts performance on reasoning-intensive benchmarks like VQA, GQA, SQA, MMBench, and MME by up to 32%, improving accuracy on tasks requiring spatial and compositional understanding.

- **Hallucination Reduction:** Enhances performance on POPE, HallusionBench, and AMBER by up to 15%, enabling TinyLLaVA 1B to achieve superior POPE performance (86.63%) compared to much larger models like LLaVA 7B (86.60%) and Qwen 2.5 VL 3B (85.9%) by improving cross-modal grounding when finetuning on the same dataset.
- **Compute Efficiency:** Parameter-free design adds negligible overhead (e.g. 0.1% inference time overhead on POPE), ideal for resource-constrained settings like single-GPU environments.
- **Plug-and-Play Simplicity:** Requires minimal changes to existing VLM architectures, enabling direct adoption of SOTA pretrained LLMs into MLLMs.

## 2 RELATED WORK

**Cross Attention-based VLM** Flamingo (Alayrac et al., 2022) is an influential Vision Language Model designed to address the alignment problem in MLLMs by introducing a cross-attention mechanism between visual and language features. Instead of projecting visual tokens directly into the language embedding space, Flamingo allows the language model to attend to the outputs of a frozen vision encoder using interleaved cross-attention layers. This design enables the model to dynamically condition on visual content when generating text, improving multimodal understanding. However, while Flamingo avoid the positional bias by introducing a novel interleaved cross-attention interface between frozen vision and language encoders, it nevertheless falls short of pure self-attention VLMs in reasoning and general language understanding. This underperformance stems from its compressed visual representations, shallow fusion layers, and inherently asymmetric information flow—text tokens attend to vision tokens, but not vice versa—hindering deep cross-modal integration. More recent self-attention architectures such as LLaVA-MORE (Liu et al., 2024a) report superior results across benchmarks like VQA, GQA, and POPE, demonstrating stronger grounding and reasoning capabilities than cross-attention models.

**Circle Rotary Position Embedding** Circle-RoPE (Wang et al., 2025) address this challenge of positional noise by proposing a novel positional encoding framework that decouples text and image token indices to enhance multimodal understanding. Circle-RoPE projects image token indices onto a 3D circular trajectory orthogonal to the text index direction, ensuring consistent RoPE distances (Per-Token Distance, PTD = 0) between text and image tokens, thus eliminating artificial positional dependencies while preserving intra-image spatial relationships through a mixed-angle circular mapping. However, while Circle-RoPE and other similar approaches offer a potential theoretical fix, altering the positional encoding of a pretrained LLM backbone is rarely feasible in practice, as it would require retraining or heavily finetuning the model at prohibitive cost. Consequently, most deployed VLMs preserve the modality-blinded RoPE of the backbone and instead seek lightweight mechanisms to restore cross-modal locality without disturbing pretrained positional priors.

**Supervised Embedding Alignment** SEA (Yin et al., 2024) is a novel token-level alignment strategy that leverages vision-language pre-trained models such as CLIP to generate fine-grained semantic labels for visual tokens through cosine similarity-based matching from a curated word list (Yin et al., 2024). These labels are then used to explicitly supervise the adapter via a contrastive loss during pre-training, guiding the transformation of visual patches into the LLM's embedding space. SEA avoids additional annotation costs, maintains inference efficiency, and improves performance across multiple benchmarks and LLM scales. Their results demonstrate that SEA enhances alignment quality and model robustness, particularly in smaller LLMs where misalignment is more severe. However, SEA completely ignore the positional noises mentioned earlier that also exacerbate the misalignment issue. Furthermore, SEA relies on the frozen CLIP projector—originally trained only for aligning global [CLS] and [EOS] embeddings—to adjust the dimensions of all image and text tokens when producing alignment labels. Since this projector was never optimized for fine-grained token-level correspondence, it may inherently introduce inaccurate supervision signals, fundamentally constraining the reliability of the learned alignment.

## 3 METHODS

### 3.1 ATTENTION ANCHOR: AUGMENTING IMAGE SEQUENCES WITH TEXT TOKENS

To enhance cross-modal alignment in vision-language models, we propose AttAnchor, a lightweight framework that strategically augments the input token sequence by inserting text tokens into the

image token stream, as detailed in Algorithm 1. For a given image encoded into $M$ tokens $\{\mathbf{I}_1, \ldots, \mathbf{I}_M\}$ via a CLIP vision encoder with a multimodal projector and a text prompt with $N$ tokens $\{\mathbf{T}_1, \ldots, \mathbf{T}_N\}$, AttAnchor computes cosine similarities between each text token and all image tokens to identify the most similar image token above a threshold $\tau_{\text{align}}$. The text token is then copied and inserted immediately after this matched image token, preserving the original relative image token order and appending the full text prompt unchanged at the end. To optimize the model for this reordered sequence, we apply LoRA (Hu et al., 2022) to fine-tune both the multimodal projector and the LLM backbone, keeping the vision encoder frozen and ensuring efficient adaptation with minimal parameter updates. This approach ensures that the transformer prioritizes content-driven connections between text and image tokens, facilitating accurate attention in deeper layers for tasks requiring visual grounding, while incurring negligible computational cost, making it highly efficient for resource-limited settings like single-GPU training. We provide a comprehensive theoretical analysis of AttAnchor's effectiveness in Appendix I, including formal proofs of attention score improvements and time complexity analysis.

---

**Algorithm 1** AttAnchor (Text-into-Image): Text Token Insertion into Image Sequence

---

**Require:** Image token embeddings $\{\mathbf{I}_1, \ldots, \mathbf{I}_M\} \subset \mathbb{R}^d$ from CLIP vision encoder, text prompt token embeddings $\{\mathbf{T}_1, \ldots, \mathbf{T}_N\} \subset \mathbb{R}^d$, cosine similarity threshold $\tau_{\text{align}}$
**Ensure:** Reordered multimodal token sequence with unmodified text prompt
1: Initialize empty sequence sequence $\leftarrow [\,]$
2: Define alignment threshold $\tau_{\text{align}} \in [0, 1]$
3: **for** $m = 1$ to $M$ **do**
4:      Append image token $I_m$ to sequence
5: **end for**
6: **for** $n = 1$ to $N$ **do**
7:      Compute cosine similarities:

$$\text{sim}(m) \leftarrow \frac{\mathbf{T}_n \cdot \mathbf{I}_m}{\|\mathbf{T}_n\| \cdot \|\mathbf{I}_m\|}, \quad \forall m \in \{1, \ldots, M\}$$

8:      Find best matching image token:

$$m^*, s^* \leftarrow \arg\max_m \text{sim}(m), \quad \text{sim}(m^*) = s^*$$

9:      **if** $s^* \geq \tau_{\text{align}}$ **then**
10:          Insert text token $T_n$ into sequence immediately after $I_{m^*}$
11:      **end if**
12: **end for**
13: **for** $n = 1$ to $N$ **do**
14:      Append text token $T_n$ to sequence
15: **end for**
16: **return** sequence

---

## 3.2 ATTANCHOR (IMAGE-INTO-TEXT): IMAGE-INTO-TEXT INSERTION

For comparison, we also proposed AttAnchor (Image-into-Text) (detailed in Appendix C), another innovative approach that inserts copied image tokens into the text sequence based on cosine similarity, prepending the full image sequence to maintain visual context. The architecture of this approach is illustrated in Appendix B. While this method could potentially leverage padding tokens as pause tokens (Goyal et al., 2023) to allow the model to process mixed-modality inputs longer, it disrupts the text prompt's contiguous structure, critical for tasks like compositional reasoning or object detection, leading to suboptimal performance in smaller models like TinyLLaVA-1B due to their limited capacity to handle fragmented linguistic context. Larger models like LLaVA-7B, with greater robustness to such disruptions, benefit from the enhanced cross-modal locality, outperforming the baseline by leveraging stronger alignment capabilities. This contrasts with our primary AttAnchor approach, which preserves text prompt causality for superior cross-modal alignment across all model scales.

## 4 EXPERIMENTS

### 4.1 EVALUATION DATASETS, METRICS AND BENCHMARKS

We evaluate our approach on 3 instruction-following datasets of varying scales across 15 metrics and benchmarks. Details are in Appendix A and Appendix D.

(1) **Token-level metrics** measure fit through negated cross-entropy loss on the validation dataset;

(2) **Sentence-level evaluation metrics** assess text quality using BLEU (Papineni et al., 2002), ME-TEOR (Banerjee & Lavie, 2005), CIDEr (Vedantam et al., 2015), ROUGE-L (Lin, 2004), and BERTScore (Zhang et al., 2019) from n-gram overlap to semantic similarity on the validation dataset;

(3) **Visual Question Answering (VQA) benchmarks** test visual reasoning on VQAv2 (Goyal et al., 2017), GQA (Hudson & Manning, 2019), ScienceQA (Lu et al., 2022), and VizWiz (Gurari et al., 2018);

(4) **Multimodal benchmarks** assess comprehensive understanding via MMBench (Lu et al., 2022) and MME (Fu et al., 2023).

(5) **Hallucination benchmarks** evaluate object hallucination reduction using POPE (Li et al., 2023b), HallusionBench (Guan et al., 2024), and AMBER (Wang et al., 2023a);

### 4.2 TRAINING CONFIGURATIONS

To better unveil the effect of the cross-modal positional bias, we employ TinyLlava 1B as our primary model for its size and simplicity, as larger VLMs can compensate this bias and obscure its real impact via deeper layers. All experiments were conducted on NVIDIA H100 GPUs with 80 GB of memory, and each run completed within 10 hours. We conducted a comprehensive exploration of hyperparameter settings to ensure robust training, and the final optimal configuration is detailed in Table 1. The training configurations for LLaVA-7B and Qwen2.5-VL-3B experiments are provided in Appendix G. To balance efficiency and coverage, we empirically determined a threshold that selects roughly 10% of the text tokens as attention anchors, which are subsequently interleaved with the image tokens. Early stopping is applied to guard against overfitting, and a sufficiently large maximum number of epochs is employed to prevent underfitting. Both the modified and the original models are trained with the same configurations for fair comparisons. Unit tests have been added to ensure setting Similarity Threshold to 1 will generate exactly the same output as the original Tinyllava model, because no cosine similarity can be greater than 1 and thus no reordering will happen. All code and necessary data will be made publicly available upon publication to ensure reproducibility.

### 4.3 FROM TOKENS TO REASONING: A COMPREHENSIVE RADAR ANALYSIS OF CROSS-MODAL ALIGNMENT

The radar chart in Figure 4.3 unveils a compelling comparison of our proposed methods—AttAnchor (Text-into-Image), AttAnchor (Image-into-Text) (image token insertion into text sequence), and the baseline (no insertion)—following fine-tuning with identical configurations. As visualized in Appendix E, AttAnchor emerges as the standout performer, its expansive radar polygon signaling robust gains across token-level precision, sentence-level coherence, hallucination reduction, and reasoning capabilities, driven by its seamless integration of text signposts that preserve prompt causality while enhancing cross-modal alignment. AttAnchor (Image-into-Text), while showing some improvement, exhibits a fragmented polygon, reflecting inconsistent alignment due to text sequence disruption, which undermines its effectiveness compared to AttAnchor in smaller models. The baseline, with its constricted shape, highlights the inherent limitations of standard VLMs in overcoming positional distortions, reinforcing the need for strategic token reordering. This analysis underscores AttAnchor (Text-into-Image)'s superiority in unlocking true content-based interactions, offering a scalable solution for resource-constrained settings.

Table 1: Cosine Similarity Training Configuration for TinyLlava Experiments

| Parameter | Value |
|---|---|
| Base Model | tiny-llava-v1-hf |
| Image Tokens | 576 |
| Training Epochs | 4 |
| Early Stopping Patience | 10,000 steps |
| Early Stopping Criteria | Validation Loss |
| Best Epoch | 2 |
| Datatype | fp16 |
| Similarity Threshold | 0.12 |
| LoRA Rank | 32 |
| LoRA Alpha | 64 |
| LoRA Dropout | 0.05 |
| Per Device Batch Size | 4 |
| Gradient Accumulation Steps | 4 |
| Learning Rate | $1 \times 10^{-4}$ |
| Learning Rate Scheduler | Cosine |
| Warmup Ratio | 0.03 |
| Weight Decay rate | 0.003 |
| Optimizer | AdamW |

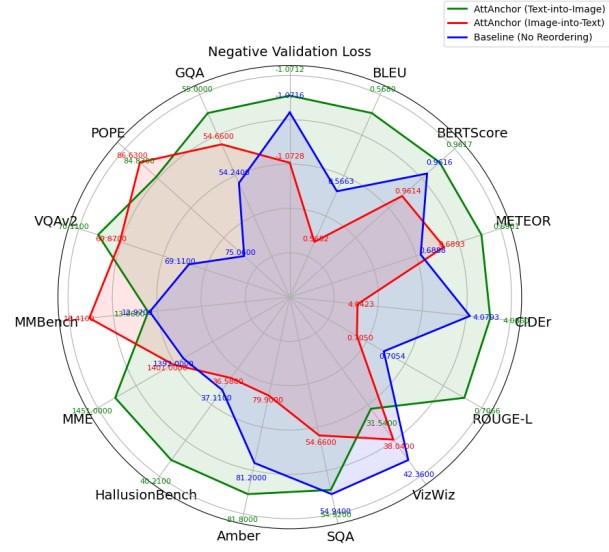

Figure 4: Tinyllava Radar Map Comparison for AttAnchor (Text-into-Image), AttAnchor (Image-into-Text), and Baseline (No Reordering).

## 4.4 ANALYZING ATTANCHOR'S TRAINING DYNAMICS

The loss plot in Figure 5 offers a compelling narrative of AttAnchor's evolution against the baseline on the LLaVA-7B model, revealing a dynamic shift in performance over training iterations. Initially, the baseline is competitive with AttAnchor in losses, leveraging the original architecture's established stability, providing a familiar starting point for the transformer. However, as training progresses, AttAnchor adapts, harnessing its text token insertions as semantic signposts to guide cross-modal attention, significantly surpassing the baseline with a steadily declining loss curve. This transition underscores AttAnchor's ability to efficiently rewire the model, transforming text cues into powerful anchors that enhance image-text alignment, a clear advantage for resource-constrained settings. Similar training dynamics are observed for Qwen2.5-VL-3B experiments, as shown in Appendix F. The detailed training configurations for these experiments are provided in Appendix G.

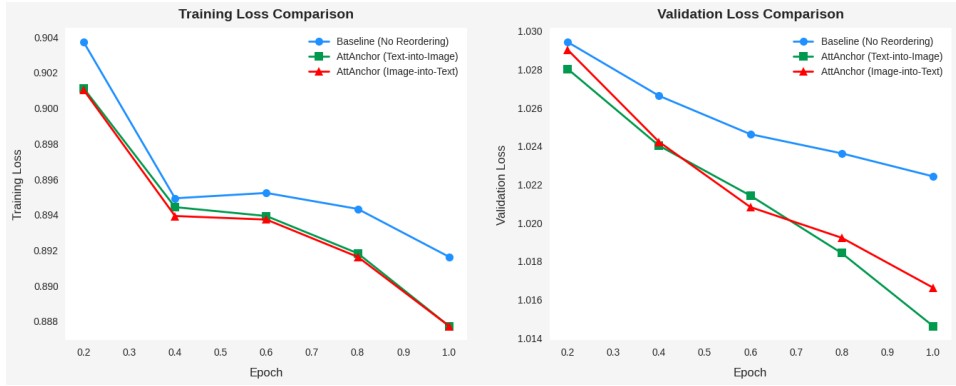

Figure 5: Llava 7B Training and Validation Loss Plots for AttAnchor (Text-into-Image), AttAnchor (Image-into-Text), and Baseline (No Reordering). All three models reach their best performance at the first epoch.

## 4.5 THRESHOLD ANALYSIS FOR CROSS-MODAL TOKEN REORDERING

Table 2 presents the impact of varying the cosine similarity threshold $\tau_{\text{align}}$ in AttAnchor (Text-into-Image), which controls the number of text tokens inserted as attention anchors to annotate and enhance the image token sequence. Lowering the threshold from 1.0 (baseline) to 0.08 increases the number of inserted text tokens, enriching cross-modal alignment by providing more semantic cues for the transformer to focus on relevant image regions. This leads to optimal performance at $\tau_{\text{align}} = 0.12$ or $0.14$, where the model outperforms the baseline across 13 out 15 metrics, as these thresholds balance the addition of informative anchors with minimal noise from less accurate similarity matches. Notably, as we lower the cosine similarity threshold, hallucination (POPE, HallusionBench, Amber) and multimodal benchmarks (MME, MMBench) show significant improvements, particularly for smaller models like TinyLLaVA-1B, due to enhanced grounding from additional anchors, which guide deeper transformer layers to prioritize content-driven interactions. Interestingly, for VizWiz, which contains lower-quality, blurry images, setting a higher threshold (e.g., $\tau_{\text{align}} = 0.14$) could achieve gains over the baseline, as lower thresholds (e.g., 0.12 or below) introduce noise by inserting text tokens based on less reliable cosine similarity scores, disrupting alignment in visually challenging settings (Gurari et al., 2018). This threshold-dependent behavior underscores the AttAnchor (Text-into-Image)'s adaptability, enabling robust performance across diverse tasks while maintaining efficiency in resource-constrained environments.

Table 2: Comparison of Tinyllava 1B performance metrics across different reordering thresholds.

| Category | Performance/Threshold | 1 (Baseline) | 0.16 | 0.14 | 0.12 | 0.10 | 0.08 |
|---|---|---|---|---|---|---|---|
| Loss | Val Loss (Negated) | -1.0716 | -1.0706 | -1.0704 | -1.0712 | -1.0706 | -1.0710 |
| Eval Metrics | BLEU | 0.5663 | 0.5654 | 0.5698 | 0.5680 | 0.5652 | 0.5670 |
| | BERTScore | 0.9616 | 0.9616 | 0.9619 | 0.9617 | 0.9615 | 0.9617 |
| | METEOR | 0.6888 | 0.6877 | 0.6917 | 0.6901 | 0.6898 | 0.6895 |
| | CIDEr | 4.0793 | 4.0665 | 4.0990 | 4.0860 | 4.0508 | 4.0781 |
| | ROUGE-L | 0.7054 | 0.7058 | 0.7080 | 0.7066 | 0.7051 | 0.7061 |
| VQA | VQAv2 | 69.11 | 69.74 | 69.64 | 70.11 | 69.77 | 70.45 |
| | SQA | 54.94 | 55.46 | 55.25 | 54.92 | 54.94 | 54.14 |
| | GQA | 54.24 | 55.26 | 54.04 | 55.00 | 54.69 | 54.95 |
| | VizWiz | 42.36 | 36.09 | 43.24 | 31.54 | 37.49 | 36.21 |
| Multimodal | MMBench | 12.97 | 15.29 | 17.10 | 13.06 | 11.86 | 15.29 |
| | MME | 1392 | 1391 | 1371 | 1452 | 1424 | 1465 |
| Hallucination | POPE | 75.06 | 61.9 | 75.4 | 84.8 | 81.7 | 85.1 |
| | Amber | 81.2 | 81.4 | 81.6 | 81.8 | 81.3 | 82.3 |
| | HallusionBench | 37.11 | 37.73 | 39.50 | 40.21 | 39.15 | 39.06 |

## 4.6 FROM DISRUPTION TO ALIGNMENT: COMPARING ATTENTION ANCHOR (TEXT-INTO-IMAGE) AND ATTANCHOR (IMAGE-INTO-TEXT) ACROSS MODEL SCALES

Table 3 compares the AttAnchor (Text-into-Image) and AttAnchor (Image-into-Text) across different model scales, revealing distinct performance trends. For smaller models like TinyLLaVA-1B, AttAnchor (Text-into-Image) improves 13/15 metrics, but AttAnchor (Image-into-Text) improves only 6/15 metrics, as inserting image tokens into the text sequence disrupts the causal semantic flow critical for coherent prompt processing, which limited-capacity models struggle to recover from. In contrast, larger models like LLaVA-7B, with greater capacity to handle mixed-modality prompts, leverage AttAnchor (Image-into-Text)'s enhanced locality to improve 11/15 metrics, demonstrating robustness to text disruptions. Conversely, AttAnchor (Text-into-Image) excels in smaller models, achieving significant gains and improving 13/15 metrics compared to improving 11/15 metrics for 7B model, particularly in hallucination benchmarks, as their fewer layers are less adept at compensating for positional biases, making text-to-image insertion's alignment benefits more pronounced. These findings highlight AttAnchor (Text-into-Image)'s superior adaptability for resource-constrained settings, where smaller models benefit most from precise cross-modal grounding. Similar experimental results and training configurations for Qwen2.5-VL-3B are provided in Appendix G and Appendix H.

Table 3: Performance Comparison: TinyLlava 1B vs Llava 7B across Different Methods

| Performance | TinyLlava 1B | | | Llava 7B | | |
|---|---|---|---|---|---|---|
| | Baseline | AttAnchor | AttAnchor (Img) | Baseline | AttAnchor | AttAnchor (Img) |
| Loss (Negated) | -1.0716 | -1.0712 | -1.0728 | -1.0224 | -1.0146 | -1.0166 |
| BLEU | 0.5663 | 0.5680 | 0.5652 | 0.5819 | 0.5828 | 0.5833 |
| BERTScore | 0.9616 | 0.9617 | 0.9614 | 0.9636 | 0.9636 | 0.9637 |
| METEOR | 0.6888 | 0.6901 | 0.6893 | 0.7021 | 0.7033 | 0.7029 |
| CIDEr | 4.0793 | 4.0860 | 4.0423 | 4.2556 | 4.2837 | 4.2825 |
| ROUGE-L | 0.7054 | 0.7066 | 0.7050 | 0.7192 | 0.7200 | 0.7203 |
| VQAv2 | 69.11 | 70.11 | 69.87 | 76.16 | 75.86 | 75.94 |
| SQA | 54.94 | 54.92 | 54.66 | 68.22 | 69.87 | 68.62 |
| GQA | 54.24 | 55.00 | 54.66 | 60.91 | 60.60 | 60.57 |
| VizWiz | 42.36 | 31.54 | 38.04 | 47.6 | 48.0 | 49.5 |
| MMBench | 12.97 | 13.06 | 16.41 | 47.65 | 47.68 | 49.39 |
| MME | 1392 | 1452 | 1401 | 1784 | 1800 | 1788 |
| POPE | 75.06 | 84.83 | 86.63 | 86.60 | 86.70 | 86.63 |
| Amber | 81.2 | 81.8 | 79.9 | 86.1 | 86.3 | 85.9 |
| HallusionBench | 37.11 | 40.21 | 36.58 | 42.07 | 40.12 | 41.63 |

## 5 CONCLUSION

In this work, we introduced AttAnchor, a novel parameter-free method for enhancing cross-modal alignment in vision-language models through strategic token reordering. By inserting text tokens near semantically similar image patches based on cosine similarity, AttAnchor creates semantic signposts that guide attention to relevant image regions, addressing the fundamental positional bias problem in multimodal transformers.

Our evaluation across 15 metrics and benchmarks demonstrates AttAnchor's effectiveness, with improvements in 13 out of 15 evaluated metrics. The method shows consistent gains across token-level, sentence-level, reasoning-level, and hallucination reduction tasks while maintaining computational efficiency and requiring minimal architectural changes.

AttAnchor highlights the importance of addressing cross-modal positional biases and opens new directions for mixed-modal token grouping research. The parameter-free, plug-and-play design makes it particularly valuable for practitioners working with limited computational resources.

## 6 LIMITATIONS AND FUTURE WORKS

**Addressing Internal Misalignment**    QwenVL experiments in Appendix H show that some VLMs suffer from internal misalignment across modalities, significantly reducing optimal similarity threshold and reliable image-text matches required by AttAnchors. Therefore, a promising direction is to integrate our method with Supervised Embedding Alignment (SEA) discussed in Section 2, which provides explicit token-level content supervision using CLIP-based semantic labels and improves content alignment. Combining explicit content alignment from SEA with our positional alignment strategy could yield a more comprehensive solution to multimodal token misalignment, further enhancing robustness and generalization in MLLMs.

**Extension to Longer Image Sequences and Multimodal LLMs**    In our experiments, number of generated image tokens is fixed at 576, we believe our approach will be more effective after increasing the number of image tokens, and the proposed approach can be naturally extended to broader multimodal large language models. In particular, future work will explore incorporating additional modalities such as audio, video, and sensor streams (e.g., IMU data), which would enable richer contextual understanding and more versatile applications. Extending our framework to these domains has the potential to further validate its flexibility and effectiveness, as well as open new directions for multimodal reasoning and interaction.

**Synergizing with Token Reduction**    Our current approach uses some redundant tokens as AttAnchors. Thus, another promising future direction is to combine AttAnchor with token reduction, leveraging its enhanced token locality and alignment to prune low-cosine-similarity image tokens, potentially improving computational efficiency and memory usage in resource-constrained settings without sacrificing performance.

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

## A  DATASET DETAILS

We evaluate our approach on three instruction-following datasets of varying scales: LLaVA-Instruct-150K ( 150k samples) (Liu et al., 2023) for primary training, llava-instruct-mix-vsft-mini ( 8k samples) (UnslothAI, 2024) for prototyping, and llava-instruct-mix-vsft ( 270k samples) (HuggingFaceH4, 2024) for additional verification. These datasets provide diverse multimodal instruction-following scenarios with varying question formats from open-ended to multiple-choice.

**LLaVA-Instruct-150K.** This dataset, containing roughly 150k samples, is the primary training corpus used in our experiments. It was also one of the key datasets employed in the original LLaVA model training (Liu et al., 2023). Unlike benchmark datasets, it contains only open-ended questions, making it particularly well-suited for avoiding overfitting to specific evaluation formats.

**llava-instruct-mix-vsft-mini.** This smaller variant comprises about 8,000 samples (UnslothAI, 2024). Each entry includes an image, a question grounded in the image, and the corresponding ground-truth answer. In addition to standard open-ended responses, the dataset contains multiple-choice and true/false questions, providing more benchmark-like evaluation formats. We primarily used this dataset for early prototyping and testing.

**llava-instruct-mix-vsft.** This larger-scale dataset, released by HuggingFaceH4, contains about 270k multimodal instruction-following samples (HuggingFaceH4, 2024). Each entry pairs an image with a natural language instruction or question and a reference answer. Compared with the mini version, it offers broader coverage of concepts and instruction types, and we used it mainly for additional verification.

# B ATTANCHOR (IMAGE-INTO-TEXT) ARCHITECTURE

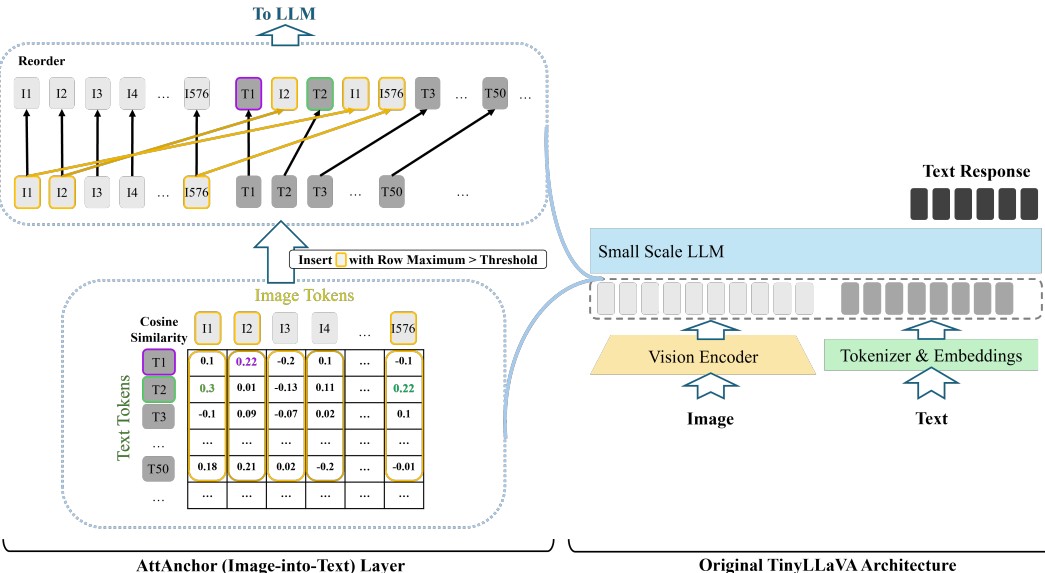

Figure 6: AttAnchor (Image-into-Text) Architecture.

Figure 6 illustrates the AttAnchor (Image-into-Text) architecture, which represents the reverse approach to our primary AttAnchor method. Unlike AttAnchor (Text-into-Image) that inserts text tokens into the image sequence, this approach inserts image tokens into the text sequence based on cosine similarity. The architecture shows how image patches are copied and strategically placed near semantically similar text tokens, creating visual anchors that guide attention within the text sequence. This method prepends the full image sequence to maintain visual context while disrupting the text prompt's contiguous structure. While this approach enhances cross-modal locality for larger models with greater capacity to handle fragmented linguistic context, it can lead to suboptimal performance in smaller models due to their limited ability to recover from text sequence disruptions.

# C AttAnchor (Image-into-Text) Algorithm: Image-into-Text Insertion

---

**Algorithm 2** AttAnchor (Image-into-Text): Image Token Insertion into Text Sequence

---

**Require:** Image token embeddings $\{\mathbf{I}_1, \ldots, \mathbf{I}_M\} \subset \mathbb{R}^d$ from CLIP, text token embeddings $\{\mathbf{T}_1, \ldots, \mathbf{T}_N\} \subset \mathbb{R}^d$, threshold $\tau_{\text{align}}$, number of pad (pause) tokens $K \geq 0$, think mode flag think_mode

**Ensure:** Reordered multimodal token sequence with optional pause tokens

1: Initialize sequence $\leftarrow [\,]$
2: **for** $m = 1$ to $M$ **do**
3:      Append $I_m$ to sequence
4: **end for**
5: **for** $n = 1$ to $N$ **do**
6:      Append $T_n$ to sequence
7: **end for**
8: **for** $m = 1$ to $M$ **do**
9:      Compute cosine similarities:

$$\text{sim}(n) \leftarrow \frac{\mathbf{I}_m \cdot \mathbf{T}_n}{\|\mathbf{I}_m\| \cdot \|\mathbf{T}_n\|}, \quad \forall n \in \{1, \ldots, N\}$$

10:      Find best match: $n^*, s^* \leftarrow \arg\max_n \text{sim}(n)$
11:      **if** $s^* \geq \tau_{\text{align}}$ **then**
12:          Insert $I_m$ after $T_{n^*}$ in sequence
13:      **end if**
14: **end for**
15: **if** think_mode **then**
16:      **for** $k = 1$ to $K$ **do**
17:          Append pause token $P_k$ to sequence with attention mask = 1, label = $-100$
18:      **end for**
19: **end if**
20: **return** sequence

---

# D Evaluation Metrics and Benchmarks Details

## D.1 Token-Level Metric

**Negative Validation Loss.** We use cross-entropy on the held-out validation set as a token-level measure of fit. For consistency in radar plots, we negate this value so that higher scores indicate better performance.

## D.2 Sentence-Level Metrics

**BLEU (Papineni et al., 2002).** BLEU computes $n$-gram precision with a brevity penalty to discourage short outputs. It is widely used but sensitive to surface overlap and less robust to paraphrasing.

**BERTScore (Zhang et al., 2019).** BERTScore compares contextual embeddings of tokens from candidate and reference texts. This makes it more robust to paraphrasing than surface metrics.

**METEOR (Banerjee & Lavie, 2005).** METEOR aligns candidate and reference texts using stemming and synonyms. It emphasizes recall and correlates better with human judgments than BLEU.

**CIDEr (Vedantam et al., 2015).** CIDEr evaluates consensus with multiple references using TF–IDF–weighted $n$-grams. It is specifically designed for image captioning tasks.

**ROUGE-L (Lin, 2004).**  ROUGE-L measures the longest common subsequence between candidate and reference. It reflects sentence-level fluency and word order.

## D.3  VQA BENCHMARKS

**VQAv2 (Goyal et al., 2017).**  VQAv2 mitigates language priors by pairing similar images that require different answers. It remains the standard dataset for visual question answering.

**ScienceQA (SQA) (Lu et al., 2022).**  ScienceQA contains multimodal multiple-choice science questions with curated explanations. It spans natural sciences, language, and social sciences.

**GQA (Hudson & Manning, 2019).**  GQA tests compositional reasoning using scene graphs derived from Visual Genome. It includes millions of real-world questions across diverse reasoning types.

**VizWiz (Gurari et al., 2018).**  VizWiz evaluates VQA models on real-world images taken by people who are blind, featuring questions about daily life challenges. It includes answerable and unanswerable questions, testing model robustness on low-quality images and accessibility scenarios.

## D.4  GENERAL MULTIMODAL BENCHMARKS

**MMBench (Lu et al., 2022).**  MMBench is a holistic benchmark that covers both perception tasks and reasoning tasks. It evaluates general multimodal understanding beyond VQA.

**MME (Fu et al., 2023).**  MME is a comprehensive benchmark that evaluates both perception and cognition abilities across 14 subtasks, ranging from basic visual perception (existence, count, position, color) to complex reasoning (commonsense, numerical calculation, code reasoning).

## D.5  HALLUCINATION BENCHMARKS

**POPE (Li et al., 2023b).**  POPE probes object hallucinations in LVLMs using polling-based binary questions. It contains random, popular, and adversarial subsets. The average accuracy of these 3 subsets is reported, and it is the easiest hallucination benchmark used in this paper.

**AMBER (Wang et al., 2023a).**  AMBER evaluates existence, attribute, and relation hallucinations. It is designed to be LLM-free and computationally efficient. It is the most comprehensive hallucination benchmark used in this paper.

**HallusionBench (Guan et al., 2024).**  HallusionBench targets entangled hallucinations and visual illusions. It provides controlled question pairs for diagnostic evaluation. It is the most difficult hallucination benchmark used in this paper.

## E  VISUALIZE TINYLLAVA ATTENTION ANCHORS (TEXT-INTO-IMAGE)

To visualize how Attention Anchors help assist the VLM through reordering. The text of each Semantic Signposts (in green) are attached to the positions of their best-match image tokens. As shown below, the text attention anchors are mostly well-aligned to the relevant image tokens.

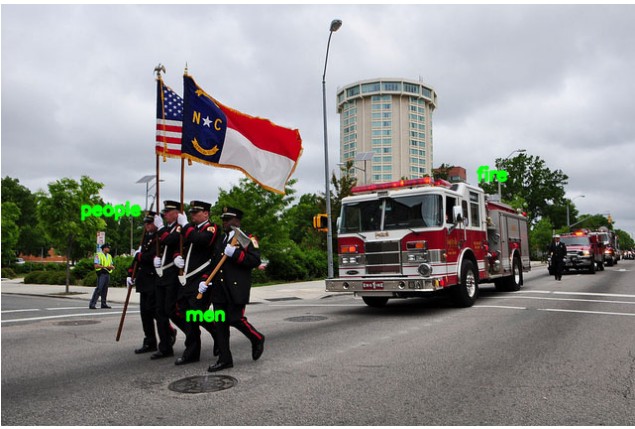

**Prompt:** Are the uniformed people in the image firemen or service men?
**Model Answer:** The uniformed people in the image are firemen.

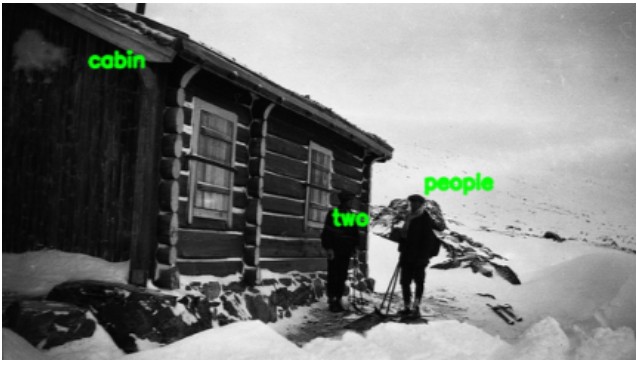

**Prompt:** Are the two people in the image inside or outside of the cabin?
**Model Answer:** The two people in the image are standing outside of the cabin.

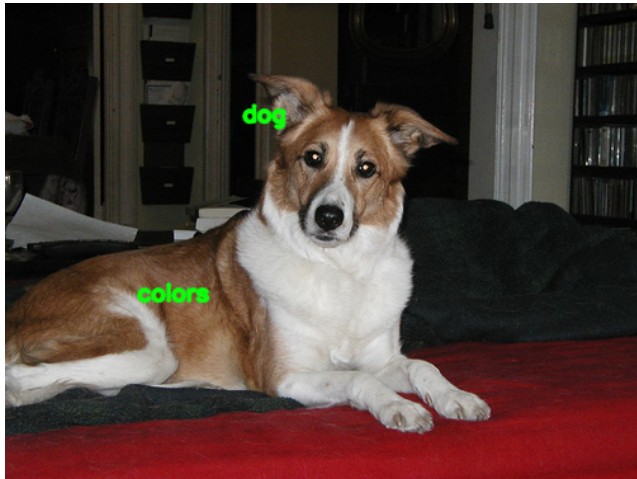

**Prompt:** What colors are the dog in the image?
**Model Answer:** The dog in the image is brown and white.

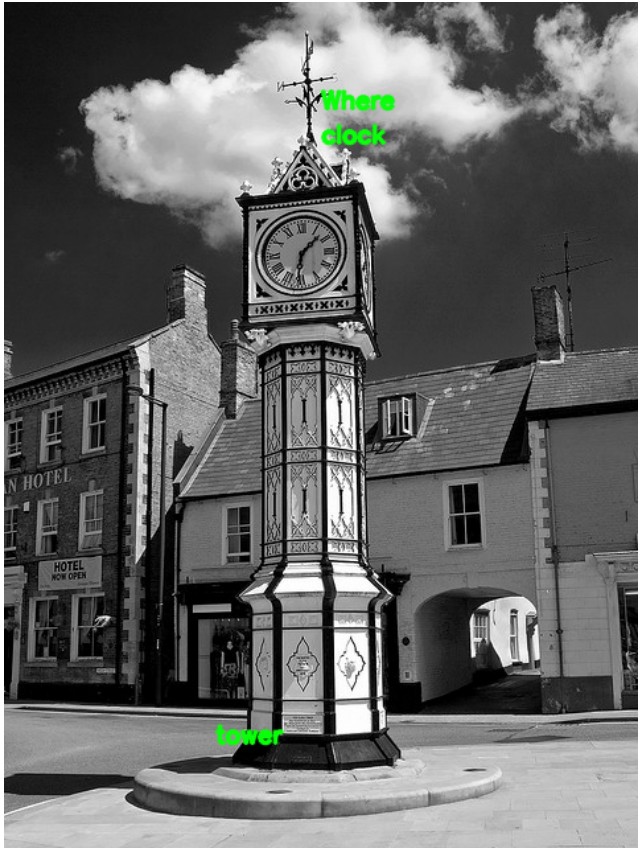

**Prompt:** Where is the clock tower located?
**Model Answer:** The clock tower is located on a street corner in a small town, standing tall in the middle of the town.

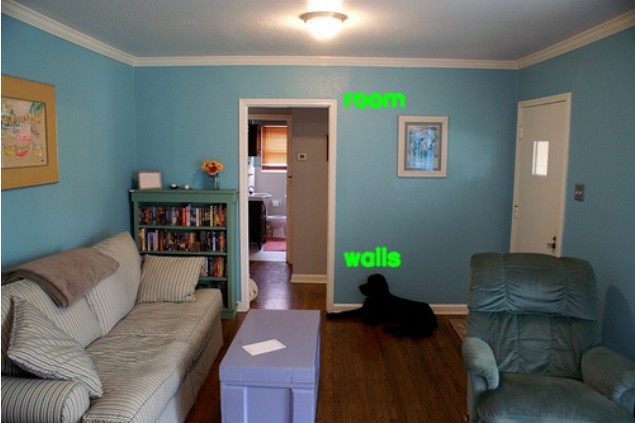

**Prompt:** What color are the walls in the room?
**Model Answer:** The walls in the room are blue.

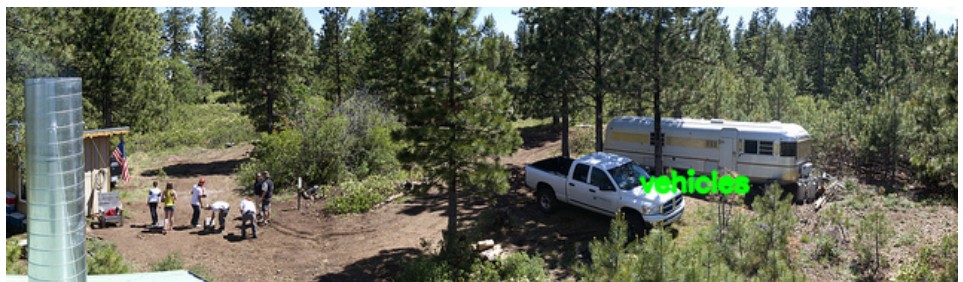

**Prompt:** What vehicles are visible in the image?
**Model Answer:** In the image, there is a white pickup truck and a camper trailer visible.

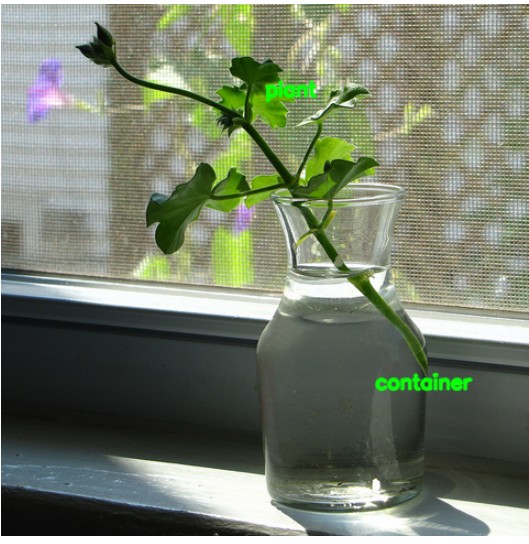

**Prompt:** What type of container holds the plant in the image?
**Model Answer:** The plant in the image is held in a glass vase.

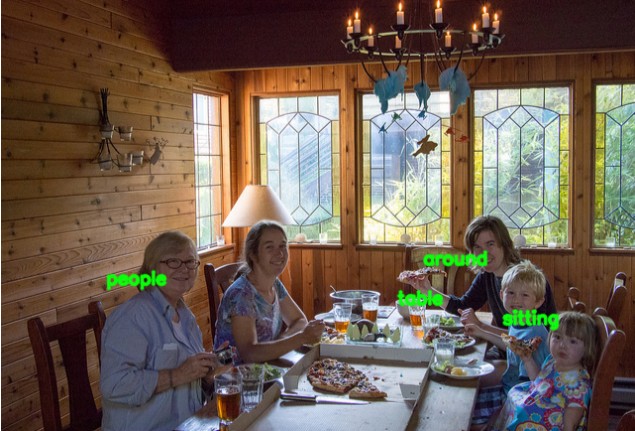

**Prompt:** What activity are the group of people sitting around the table doing?
**Model Answer:** The people in the image are sitting around a table with pizza and drinks, enjoying a meal together.

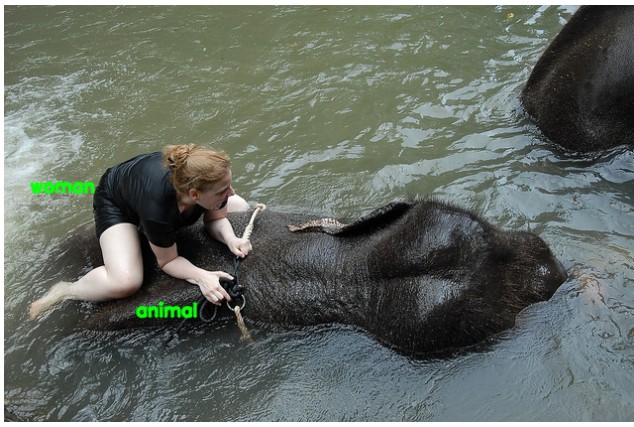

**Prompt:** What animal is the woman riding in the image?
**Model Answer:** The woman is riding an elephant in the image.

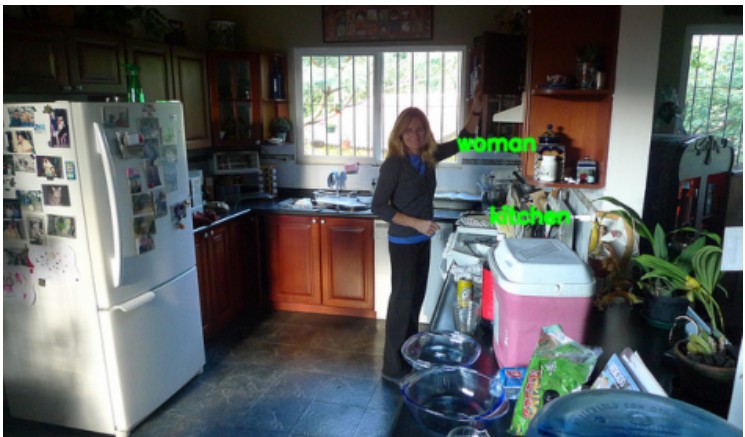

**Prompt:** Is the woman in the kitchen smiling?
**Model Answer:** Yes, the woman in the kitchen is smiling.

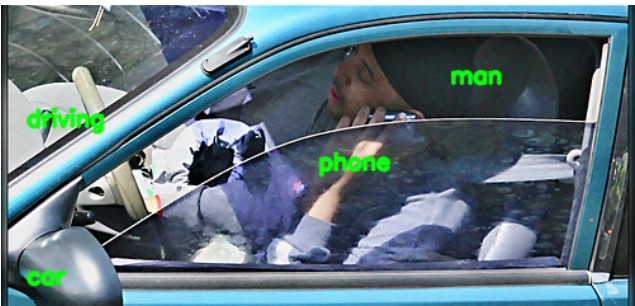

**Prompt:** Is the young man sitting or driving the car while using his cell phone?
**Model Answer:** The young man is sitting in the car while talking on his cell phone.

# F  QWEN2.5-VL-3B TRAINING DYNAMICS

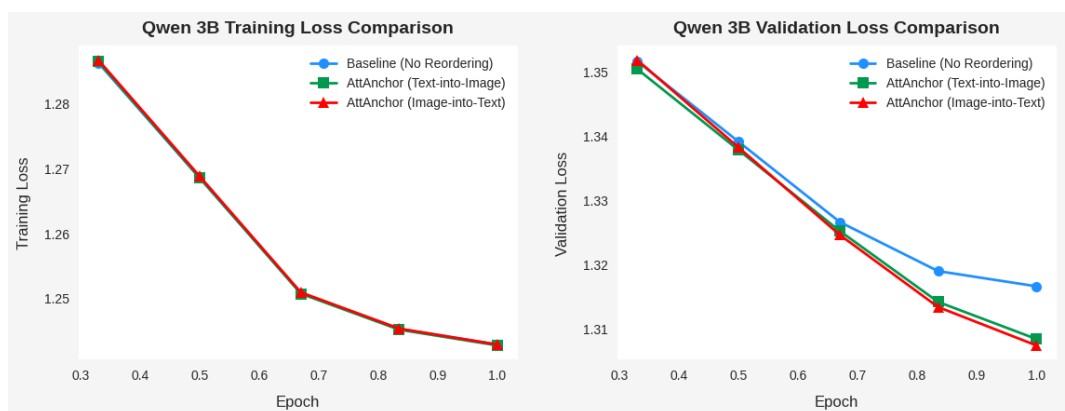

Figure 7: Qwen2.5-VL-3B Training and Validation Loss Plots for AttAnchor (Text-into-Image), AttAnchor (Image-into-Text), and Baseline (No Reordering).

The loss plots in Figure 7 demonstrate similar training dynamics for the Qwen2.5-VL-3B model, validating the consistency of AttAnchor's performance across different model architectures. Both AttAnchor (Text-into-Image) and AttAnchor (Image-into-Text) show improved validation loss trajectories compared to the baseline, with AttAnchor (Image-into-Text) achieving the lowest final validation loss. The training loss curves remain similar across all methods, indicating that the improvements stem from better generalization rather than training optimization differences.

# G  TRAINING CONFIGURATIONS FOR LLAVA-7B AND QWEN2.5-VL-3B ATTANCHOR EXPERIMENTS

| Parameter | Value |
|---|---|
| Base Model | Llava-1.5-7b-hf |
| Image Tokens | 576 |
| Training Epochs | 2 |
| Early Stopping Patience | 5,000 steps |
| Early Stopping Criteria | Validation Loss |
| Best Epoch | 1 |
| Datatype | `fp16` |
| Similarity Threshold | 0.12 |
| LoRA Rank | 8 |
| LoRA Alpha | 16 |
| LoRA Dropout | 0.05 |
| Per Device Train Batch Size | 8 |
| Gradient Accumulation Steps | 4 |
| Learning Rate | $2 \times 10^{-4}$ |
| Learning Rate Scheduler | Cosine |
| Warmup Ratio | 0.03 |
| Weight Decay rate | 0.00 |
| Optimizer | `paged_adamw_8bit` |

Table 4: Training Configuration for LLaVA-7B AttAnchor Experiments

| Parameter | Value |
|---|---|
| Base Model | Qwen2.5-VL-3B-Instruct |
| Image Tokens | 576 |
| Training Epochs | 2 |
| Early Stopping Patience | 4,500 steps |
| Early Stopping Criteria | Validation Loss |
| Best Epoch | 2 |
| Datatype | `bf16` |
| Similarity Threshold | 0.08 |
| LoRA Rank | 32 |
| LoRA Alpha | 64 |
| LoRA Dropout | 0.05 |
| Per Device Train Batch Size | 8 |
| Gradient Accumulation Steps | 4 |
| Learning Rate | $1 \times 10^{-4}$ |
| Learning Rate Scheduler | Cosine |
| Warmup Ratio | 0.1 |
| Weight Decay rate | 0.03 |
| Optimizer | `paged_adamw_8bit` |

Table 5: Training Configuration for Qwen2.5-VL-3B AttAnchor Experiments

## H   QWEN2.5-VL-3B RESULTS: LESSONS FROM A MISALIGNED VLM

Table 6: Performance Comparison: TinyLlava 1B vs QwenVL 3B across Different Methods

| Performance | TinyLlava 1B | | | QwenVL 3B | | |
|---|---|---|---|---|---|---|
| | Baseline | AttAnchor | AttAnchor (Img) | Baseline | AttAnchor | AttAnchor (Img) |
| Loss (Negated) | -1.0716 | -1.0712 | -1.0728 | -1.2928 | -1.2924 | -1.2920 |
| BLEU | 0.5663 | 0.5680 | 0.5652 | 0.5738 | 0.5726 | 0.5754 |
| BERTScore | 0.9616 | 0.9617 | 0.9614 | 0.9623 | 0.9624 | 0.9624 |
| METEOR | 0.6888 | 0.6901 | 0.6893 | 0.6961 | 0.6964 | 0.6967 |
| CIDEr | 4.0793 | 4.0860 | 4.0423 | 4.2122 | 4.1945 | 4.2356 |
| ROUGE-L | 0.7054 | 0.7066 | 0.7050 | 0.7120 | 0.7126 | 0.7130 |
| VQAv2 | 69.11 | 70.11 | 69.87 | 78.93 | 78.42 | 78.65 |
| SQA | 54.94 | 54.92 | 54.66 | 67.79 | 68.26 | 68.05 |
| GQA | 54.24 | 55.00 | 54.66 | 56.42 | 55.87 | 55.84 |
| VizWiz | 42.36 | 31.54 | 38.04 | 58.73 | 58.32 | 60.01 |
| MMBench | 12.97 | 13.06 | 16.41 | 75.94 | 76.38 | 75.43 |
| MME | 1392 | 1452 | 1401 | 1978 | 2002 | 2022 |
| POPE | 75.06 | 84.83 | 86.63 | 85.90 | 86.60 | 85.83 |
| Amber | 81.2 | 81.8 | 79.9 | 87.8 | 87.9 | 87.7 |
| HallusionBench | 37.11 | 40.21 | 36.58 | 53.41 | 52.79 | 51.37 |

The AttAnchor approach yields similar but smaller performance gain on Qwen-3B than other models, where QwenVL uses pooling to compress vision token representations, achieving improvements over the baseline in only 9/15 metrics, compared to 13/15 for TinyLLaVA-1B and 11/15 for LLaVA-7B. This reduced effectiveness results from QwenVL's internal pooling mechanism after its multimodal projector, which diminishes the granularity of image token embeddings, leading to significantly lower maximum cosine similarity scores and optimal Similarity Threshold (0.07–0.08) compared to 0.12–0.14 for TinyLLaVA and Llava on the same dataset, thus hindering reliable text-image matches for attention anchor insertion. This finding suggests that models with misaligned vision-language embeddings, like QwenVL, could potentially benefit from first using pre-alignment techniques such as Supervised Embedding Alignment (SEA) Yin et al. (2024) to strengthen embedding correspondence before applying AttAnchor, offering a novel pathway to further enhance cross-modal alignment in the future.

# I   THEORETICAL ANALYSIS OF ATTANCHOR

## I.1   ATTENTION SCORE IMPROVEMENT ANALYSIS

The effectiveness of AttAnchor can be understood through the lens of attention score distributions in transformer architectures. In standard VLMs, the attention score between text token $T_i$ and image token $I_j$ is computed as:

$$A_{ij} = \frac{\exp(Q_i K_j^T / \sqrt{d})}{\sum_{k=1}^{M+N} \exp(Q_i K_k^T / \sqrt{d})} \tag{1}$$

where $Q_i$ and $K_j$ are query and key vectors respectively, $d$ is the embedding dimension, and $M + N$ is the total sequence length. The positional encoding in RoPE introduces a bias term that penalizes distant tokens:

$$A_{ij} = \frac{\exp((Q_i K_j^T + P_{ij}) / \sqrt{d})}{\sum_{k=1}^{M+N} \exp((Q_i K_k^T + P_{ik}) / \sqrt{d})} \tag{2}$$

where $P_{ij}$ represents the positional bias. For RoPE, this bias decreases with distance: $P_{ij} < P_{ik}$ when $|i - j| > |i - k|$ (assuming $P$ is negative for larger distances).

**Theorem 1 (Attention Score Improvement):** Let $T_i$ be a text token and $I_j$ be its semantically most similar image token. If a copy of $T_i$ is inserted immediately after $I_j$ in the sequence, the attention score $A_{ij}$ increases by a factor of at least $\exp(\Delta P / \sqrt{d})$, where $\Delta P = P_{ij}^{\text{attanchor}} - P_{ij}^{\text{original}} > 0$.

**Proof:** The original attention score is:

$$A_{ij}^{\text{original}} = \frac{\exp((Q_i K_j^T + P_{ij}^{\text{original}}) / \sqrt{d})}{\sum_k \exp((Q_i K_k^T + P_{ik}^{\text{original}}) / \sqrt{d})}$$

After AttAnchor insertion (adding a copy), the distance between the copied $T_i$ and $I_j$ decreases, so $P_{ij}^{\text{attanchor}} > P_{ij}^{\text{original}}$. The new attention score becomes:

$$A_{ij}^{\text{attanchor}} = \frac{\exp((Q_i K_j^T + P_{ij}^{\text{attanchor}}) / \sqrt{d})}{\sum_k \exp((Q_i K_k^T + P_{ik}^{\text{attanchor}}) / \sqrt{d}) + \exp((Q_i K_{\text{copy}}^T + P_{i,\text{copy}}^{\text{attanchor}}) / \sqrt{d})}$$

Let $S = \sum_k \exp((Q_i K_k^T + P_{ik}^{\text{original}}) / \sqrt{d})$ be the original denominator. The new denominator is $S' = S + \Delta S$, where $\Delta S$ includes the added copy term and minor shifts in $P_{ik}$ for other $k$ (due to sequence lengthening by 1).

Since the copy has the same embedding as $T_i$, its contribution is bounded: $\exp((Q_i K_{\text{copy}}^T + P_{i,\text{copy}}^{\text{attanchor}}) / \sqrt{d}) \leq \max_k \exp((Q_i K_k^T + P_{ik}^{\text{original}}) / \sqrt{d})$. Assuming bounded keys ($|K| \leq B$) and large $M + N$ (typical in VLMs, e.g., 576+50=626), the relative change $|\Delta S / S| \leq 1/(M + N) \ll 1$.

Thus, the ratio $r = A_{ij}^{\text{attanchor}} / A_{ij}^{\text{original}} = [\exp(\Delta P / \sqrt{d})] \cdot [S/S']$.

Since $S' = S + \Delta S$ and $|\Delta S| \leq \epsilon S$ for small $\epsilon = O(1/(M + N))$, we have $1/(1 + \epsilon) \leq S/S' \leq 1/(1 - \epsilon)$, implying $r \geq \exp(\Delta P / \sqrt{d})/(1 + \epsilon) \approx \exp(\Delta P / \sqrt{d})(1 - \epsilon) > \exp(\Delta P / \sqrt{d}) - \delta$, where $\delta$ is negligible for large sequences.

## I.2   INFORMATION-THEORETIC ANALYSIS

The cross-modal alignment problem can be framed as maximizing mutual information between text and image representations. Let $X = \{I_1, \ldots, I_M\}$ be the image tokens and $Y = \{T_1, \ldots, T_N\}$ be the text tokens.

**Definition 1 (Cross-Modal Mutual Information):** The mutual information between image and text representations is:

$$I(X;Y) = H(X) - H(X|Y) = \sum_{i,j} p(x_i, y_j) \log \frac{p(x_i, y_j)}{p(x_i)p(y_j)} \tag{3}$$

**Theorem 2 (Local Information Enhancement):** AttAnchor increases the local mutual information $I(X_{local}; Y_{anchor})$ while approximately preserving the global mutual information $I(X;Y)$, where $X_{local}$ represents image tokens near inserted text anchors.

**Proof:** Consider the joint distribution $p(x_i, y_j)$ over embeddings. AttAnchor inserts copies $y_{anchor}$ near semantically similar $x_{local}$, where similarity is measured by cosine $> \tau$, implying higher $p(x_{local}, y_{anchor})$.

The local MI is $I(X_{local}; Y_{anchor}) = \sum_{(i,j) \in local} p(x_i, y_j) \log \frac{p(x_i, y_j)}{p(x_i)p(y_j)}$.

Pre-insertion: This sum is over distant pairs, where $p(x_i, y_j)$ is diluted by positional noise, reducing the log term.

Post-insertion: Insertions create new local pairs with elevated $p(x_{local}, y_{anchor})$ due to reduced distance (higher attention, per Theorem 1), increasing the sum by $\Delta I > 0$.

For global $I(X;Y)$: The insertion adds redundant $y_{anchor}$ (copies of existing $Y$), so the marginals $p(x_i)$ and $p(y_j)$ change minimally (by $O(1/N)$). The entropy $H(X)$ remains unchanged ($X$ unmodified), and $H(X|Y)$ decreases slightly due to better conditioning on anchors, but the net $\Delta I(X;Y) \approx 0$ since redundancy doesn't add new information. Formally, by data processing inequality, augmenting $Y$ with copies bounds $|\Delta I| \leq H(Y_{anchor}) \ll H(Y)$ for few insertions.

I.3  OPTIMAL THRESHOLD SELECTION

The threshold $\tau_{\text{align}}$ can be analyzed through the lens of precision-recall trade-offs in anchor selection.

**Definition 3 (Anchor Selection Precision and Recall):** Let $S$ be the set of true semantic correspondences (truly related image-text pairs) and $\hat{S}(\tau)$ be the set of predicted correspondences based on cosine similarity $> \tau$. Define:

$$\text{Precision}(\tau) = \frac{|S \cap \hat{S}(\tau)|}{|\hat{S}(\tau)|} \tag{4}$$

$$\text{Recall}(\tau) = \frac{|S \cap \hat{S}(\tau)|}{|S|} \tag{5}$$

**Theorem 4 (Optimal Threshold):** The optimal threshold $\tau^*$ maximizes the F1-score:

$$\tau^* = \arg\max_{\tau} \frac{2 \cdot \text{Precision}(\tau) \cdot \text{Recall}(\tau)}{\text{Precision}(\tau) + \text{Recall}(\tau)} \tag{6}$$

**Corollary 1:** For models with higher-quality embeddings (higher maximum cosine similarities), the optimal threshold $\tau^*$ is higher, explaining why different models require different thresholds.

**Proof:** Let $f(c) = p_{\text{true}} f_{\text{true}}(c) + (1 - p_{\text{true}}) f_{\text{false}}(c)$, where $f(c)$ represents the overall probability density function of cosine similarities across all possible text-image token pairs, combining the contributions from true correspondences and false ones; $f_{\text{true}}(c)$ and $f_{\text{false}}(c)$ are probability density functions of cosine similarities for true and false correspondences respectively, modeled as normal distributions with $f_{\text{true}}(c) \sim N(\mu_{\text{true}}, \sigma^2)$, $f_{\text{false}}(c) \sim N(\mu_{\text{false}}, \sigma^2)$, and $\mu_{\text{true}} > \mu_{\text{false}}$, with $p_{\text{true}}$ as the prior probability of a true correspondence. Define Precision = $\frac{\int_\tau^1 p_{\text{true}} f_{\text{true}}(c) dc}{\int_\tau^1 f(c) dc}$ and Recall = $\frac{\int_\tau^1 f_{\text{true}}(c) dc}{\int_0^1 f_{\text{true}}(c) dc}$. As $\mu_{\text{true}}$ (reflecting the maximum similarity) rises, $f_{\text{true}}(c)$ shifts right, delaying the

drop in Recall (since more true pairs remain above $\tau$) and reducing the overlap with $f_{\text{false}}(c)$ in the denominator of Precision, thus shifting the F1 peak (where $dF1/d\tau = 0$) to a higher $\tau^*$, as the optimal balance favors a higher threshold to maintain both high Precision and Recall.

### I.4 COMPUTATIONAL COMPLEXITY ANALYSIS

**Theorem 5 (Complexity Bounds):** The computational complexity of AttAnchor is $O(N \cdot M \cdot d)$ for similarity computation and $O(N \cdot M)$ for token insertion, where $N$ is the number of text tokens, $M$ is the number of image tokens, and $d$ is the embedding dimension.

**Proof:** Computing cosine similarity between all text-image pairs requires $N \cdot M$ dot products, each taking $O(d)$ time. Token insertion requires $O(N \cdot M)$ operations in the worst case. The total complexity is $O(N \cdot M \cdot d + N \cdot M) = O(N \cdot M \cdot d)$.

**Corollary 2:** The additional computational overhead is **negligible** compared to the $O((M+N)^2 \cdot d)$ complexity of self-attention, especially when $N \ll M$ (typical in VLM scenarios).

### I.5 THEORETICAL LIMITATIONS

**Theorem 6 (Failure Conditions):** AttAnchor fails to improve performance when:

1. The cosine similarity between text and image tokens is uniformly low (below threshold)

2. The semantic similarity is not well-captured by cosine similarity in the embedding space

3. The model's attention mechanism is already well-calibrated for cross-modal alignment

**Proof:** For (1): If $\max_{i,j} \cos(T_i, I_j) < \tau$, no insertions occur, so the sequence is unchanged and $\Delta A_{ij} = 0$ for all pairs.

For (2): If cosine does not correlate with semantics (e.g., due to poor projector), insertions create misleading local pairs, increasing noise in attention (negative $\Delta I$ per Theorem 2) and degrading loss.

For (3): If pre-insertion $A_{ij}$ already maximizes semantic alignment (e.g., via other mechanisms), added anchors introduce redundancy, potentially increasing sequence length without benefit, leading to minor overhead without gains.

## J UNSUCCESSFUL ATTEMPTS

In our pursuit of optimal cross-modal token alignment, we explored several alternative approaches that, while theoretically promising, proved unsuccessful in practice. We document these attempts here to provide valuable insights for future research and help others avoid similar pitfalls.

### J.1 LEARNING OPTIMAL TOKEN REORDERING WITH GUMBEL SOFTMAX

Our first unsuccessful attempt involved developing a learnable token reordering mechanism using the Gumbel Softmax algorithm (Jang et al., 2017; Mena et al., 2018) to learn optimal permutation matrices for cross-modal token alignment. The approach aimed to train a transformer-based reordering module that could dynamically rearrange tokens based on learned attention patterns.

**Methodology:** We designed a neural network that takes the concatenated image and text token embeddings as input and outputs a soft permutation matrix $P \in \mathbb{R}^{(M+N) \times (M+N)}$, where $M$ and $N$ are the number of image and text tokens respectively. The Gumbel Softmax was used to approximate discrete permutations during training, allowing gradient-based optimization of the reordering strategy.

**Challenges and Failures:** Despite the theoretical appeal of learning optimal reordering strategies, this approach faced several critical limitations:

- **Computational Complexity:** The permutation matrix scales quadratically with sequence length ($O((M + N)^2)$), making it computationally prohibitive for typical VLM sequences of 600+ to-

kens. The memory requirements for storing and computing gradients through such large matrices exceeded available GPU memory.

- **Training Instability:** The Gumbel Softmax approximation, while differentiable, introduced significant training instability. The temperature parameter required careful annealing schedules, and the discrete nature of permutations made convergence extremely sensitive to hyperparameter choices.
- **Local Optima:** The optimization landscape for learning permutation matrices proved highly non-convex, with numerous local optima that prevented the model from discovering meaningful reordering patterns. The learned permutations often collapsed to trivial solutions or failed to capture semantic relationships.

These challenges led us to abandon this approach in favor of the simpler, more robust cosine similarity-based method that forms the core of AttAnchor.

### J.2 MODALITY-AWARE RoPE POSITIONAL ENCODING

Our second unsuccessful attempt focused on developing a novel modality-aware variant of RoPE (Rotary Position Embedding) (Su et al., 2024) that would explicitly account for the different nature of image and text tokens in the positional encoding scheme.

**Methodology:** We designed a dual-indexing RoPE system where each token receives two positional indices: (1) a *local position index* representing its position within its modality (e.g., position within an image sequence or within a text sentence), and (2) a *cross-modal position index* representing its position across modalities (e.g., whether it belongs to the first input modality, second input modality, etc.). When computing attention between tokens from different modalities, we used the cross-modal position indices to reduce the effective distance penalty, thereby encouraging stronger cross-modal attention while preserving intra-modal positional relationships.

**Challenges and Failures:** This approach encountered a fundamental limitation that highlights the importance of preserving pretrained knowledge:

- **Catastrophic Forgetting:** Even modifying the positional encoding of just the first layer of the LLM backbone led to significant performance degradation. The pretrained language model had learned strong positional priors from its training data, and altering these encodings disrupted the carefully learned representations, causing catastrophic forgetting of the model's linguistic capabilities.
- **Training Instability:** The modified positional encodings created training instabilities, with loss curves showing erratic behavior and frequent divergence. The model struggled to adapt to the new encoding scheme while maintaining its language understanding capabilities.
- **Architectural Constraints:** The approach required significant modifications to the transformer architecture, making it incompatible with existing pretrained models and requiring full retraining from scratch, which is computationally prohibitive for large-scale VLMs.

This experience reinforced our understanding that any successful approach must work within the constraints of existing pretrained models, leading us to develop AttAnchor as a parameter-free, plug-and-play solution that preserves the original model architecture.

### J.3 LESSONS LEARNED

These unsuccessful attempts provided valuable insights that shaped our final approach:

1. **Simplicity over Complexity:** Complex learnable mechanisms often introduce more problems than they solve, especially when dealing with long sequences and pretrained models. Simple, rule-based approaches like cosine similarity can be more robust and effective.

2. **Preserve Pretrained Knowledge:** Any modification to core model components (like positional encodings) risks catastrophic forgetting. Successful approaches should work with existing architectures rather than against them.

3. **Computational Efficiency Matters:** Methods that scale poorly with sequence length are impractical for real-world VLM applications. The quadratic complexity of permutation matrices made our first approach infeasible.

4. **Generalization is Key:** Approaches that work well on small examples but fail to generalize are of limited practical value. AttAnchor's rule-based approach provides consistent improvements across diverse scenarios.

These lessons ultimately led us to develop AttAnchor as a simple, efficient, and effective solution that addresses the core problem of cross-modal token alignment without the pitfalls encountered in these alternative approaches.

## K   LLM USAGE

**Yes, to aid or polish writing.** Large language models were used to assist with writing and polishing various sections of this paper, including correcting typos and grammar errors, improving clarity, coherence, and academic tone across the abstract, introduction, related work, and conclusion sections. The models helped refine the presentation while ensuring all technical content remained accurate and faithful to the original research findings. All substantive technical contributions, experimental results, and methodological details were conceived and developed by the human authors.

**Yes, for retrieval and discovery (e.g., finding related work).** LLMs were employed to help identify and discover relevant related work in the field of vision-language models, cross-modal alignment, and attention mechanisms. The models assisted in finding recent papers and understanding connections between different research directions, which informed our literature review and positioning of our work within the broader research landscape. However, all final decisions about which papers to include and how to position our work were made by the human authors based on their domain expertise.

