# OpenReview forum: "AttAnchor: Guiding Cross-Modal Token Alignment in VLMs with Attention Anchors"
_ICLR.cc/2026/Conference — ICLR 2026 Conference Desk Rejected Submission_

### Official Review · Reviewer_9uGi · 2025-10-27

**Soundness:** 1
**Presentation:** 1
**Contribution:** 2
**Rating:** 2
**Confidence:** 4

**Summary:**

This paper investigates the connection between different modality tokens with similar semantics in VLMs, and introduces AttAnchor, an approach to group semantically similar tokens across modalities through reordering, to mitigate the influence of RoPE that can weaken connection between distant tokens.

**Strengths:**

* The problem this paper focuses on is interesting. Semantic correspondence between tokens of different modalities within sequences is under-explored.
* The perspective that the connection between distant tokens from different modalities would be weakened due to the position embeddings sounds reasonable

**Weaknesses:**

* The paper does not provide sufficient discussion about the method design. Tokens from different modalities usually do not have one-to-one semantic correspondences,  so why copying a token and placing it near its most similar token from another modality is a reasonable approach?
* The benefit of AttAnchor shown from the results seems marginal. And since the experiments are conducted mainly on TinyLLaVA-1B, it’s not sufficient to validate the effectiveness of AttAnchor.
* The writing of the paper can be improved. For example, the Related Work section fails to provide a clear research background, with only a few studies being discussed.

**Questions:**

Have the authors analyzed the attentions between semantically similar tokens across different modalities in existing VLMs? Probably that current VLMs can already capture the cross-modality semantic connections well.

---

### Official Review · Reviewer_Vh2N · 2025-10-27

**Soundness:** 2
**Presentation:** 2
**Contribution:** 2
**Rating:** 2
**Confidence:** 4

**Summary:**

This paper introduces AttAnchor, a parameter-free framework designed to improve cross-modal alignment in Vision-Language Models. The core idea is to address the performance degradation caused by long-distance dependencies between related image and text tokens, which arise from simple sequence concatenation. The proposed method calculates the cosine similarity between text and image token embeddings. It then copies text tokens and inserts them immediately after their most semantically similar image tokens, creating "semantic signposts." The authors claim this reordering enhances the model's ability to focus on relevant image regions, thereby improving performance on various tasks like VQA and reducing hallucinations, all with negligible computational overhead.

**Strengths:**

1. The paper addresses a genuine and significant issue in modern VLM architectures—the cross-modal alignment challenge exacerbated by modality-blind positional encodings.
2. The proposed method is simple to understand, parameter-free, and computationally cheap, making it an attractive plug-and-play solution, especially for resource-constrained environments.
3. The experiments show consistent improvements over the baseline (no reordering) across a majority of the 15 reported metrics, suggesting that the heuristic has some merit.

**Weaknesses:**

1. Lack of Novelty: The central idea of rearranging or grouping tokens based on semantic similarity is not new. The concept has been explored in various forms, such as token clustering for vision transformers (e.g., Huang et al., 2024, mentioned by the authors) and sparse attention mechanisms. AttAnchor applies this concept across modalities using a straightforward cosine similarity heuristic. While the specific "copy-and-insert" mechanism is an implementation detail, the underlying principle is an intuitive and incremental step rather than a novel contribution. The work feels more like an application of a known technique than a new, foundational method.

2. Insufficient Ablation Studies: The paper's analysis of its own method is too shallow. The primary ablation is on the similarity threshold τ_align (Table 2). This is insufficient to build a strong understanding of why the method works and what its limitations are. Key questions remain unanswered:
   - What is the impact of the similarity metric? Would L2 distance or a learned metric perform differently?
   - What is the effect of the insertion strategy? For instance, what happens if tokens are moved instead of copied, or if multiple text tokens are anchored to a single image region?
   - How does performance change as the number of inserted anchors (e.g., top-k tokens vs. threshold) varies? Without a more comprehensive ablation, the paper reads like a demonstration of a single heuristic rather than a thorough scientific investigation.

3. Weak Experimental Comparison and Missing Baselines: The experimental evaluation is not convincing because it is almost exclusively compared against a weak baseline: the original model with no reordering. This demonstrates that doing something is better than doing nothing, but it fails to position AttAnchor relative to other solutions for the same problem. The paper should have included comparisons against more recent and relevant baselines that also aim to improve cross-modal grounding.

4. Superficial Method Description and Lack of Theoretical Depth: The description of the method is overly simplistic. While simplicity is a strength, the paper fails to discuss the deeper implications or potential failure modes of its design. For example, it does not address how the disruption of the image patch sequence's contiguity by text token insertion affects the model's understanding of spatial relationships. The theoretical analysis provided in Appendix I, while present, feels like a post-hoc justification for a simple heuristic. It formalizes the well-understood idea that reducing distance between tokens mitigates the penalty from RoPE but offers no profound insights into why this specific mechanism is optimal or how it interacts with the complex learned representations within the transformer. The theory does not feel like it drives the method's design but rather describes its most obvious effect.

**Questions:**

See weakness

---

### Official Review · Reviewer_bpKR · 2025-10-29

**Soundness:** 1
**Presentation:** 3
**Contribution:** 2
**Rating:** 2
**Confidence:** 4

**Summary:**

This paper introduces AttAnchor, a parameter-free architectural framework designed to enhance cross-modal alignment in VLMs by strategically reordering tokens. The core mechanism, AttAnchor (Text-into-Image), involves computing cosine similarities between text tokens and image patches to identify and insert semantically similar text tokens after their best-matching image counterparts. This process aims to guide the model's attention to relevant image regions, thereby improving cross-modal locality, reducing hallucinations, and boosting performance on various VLM tasks without disrupting the text prompt's semantic flow.

**Strengths:**

*   **Novelty in Mixed-Modal Token Grouping:** The paper introduces a novel approach to mixed-modal token grouping, which explores explicit cross-modal grouping where text and image tokens are clustered jointly. This departs from prior work that largely restricted token grouping to the visual modality or relied on additional fine-grained losses.
*   **Parameter-Free and Plug-and-Play Design:** AttAnchor's parameter-free nature and its ability to integrate with existing VLM architectures with minimal changes make it a highly practical and "plug-and-play" solution. This is a significant strength, especially for practitioners working with limited computational resources, as it avoids the prohibitive costs of retraining LLM backbones or introducing complex new modules.

**Weaknesses:**

*   **Limitations of Cosine Similarity as a Semantic Measure:** While cosine similarity is used as the basis for semantic alignment, its effectiveness as a sole metric for capturing complex semantic relationships between text and image tokens can be limited. The paper does not adequately validate the efficacy of this semantic correlation measure when text content quality is poor (e.g., verbose or image-irrelevant content). In such scenarios, low-quality text tokens might be incorrectly matched with image patches, potentially introducing noise and degrading alignment rather than enhancing it. This raises concerns about the robustness of AttAnchor to variations in input text quality, which is a common challenge in real-world applications.
*   **Sensitivity of the $T_{align}$ Threshold:** The paper acknowledges that the performance of AttAnchor (Text-into-Image) is dependent on the `Similarity Threshold`, $T_{align}$. From Table 2, it is evident that the method is highly sensitive to the chosen $T_{align}$ value. For instance, while a threshold of 0.12 or 0.14 shows optimal performance for many metrics, there isn't a single configuration that consistently yields significant improvements across *all* tasks. For example, VizWiz performance actually decreases with lower thresholds (0.12 or below) compared to the baseline, suggesting a trade-off. This sensitivity makes it challenging to identify a universally optimal configuration, which could complicate its practical deployment and necessitate extensive tuning for different datasets and tasks.
*   **Concerns Regarding Scalability to Stronger Models:** The paper primarily showcases gains on smaller models like TinyLLaVA-1B, which inherently struggle with positional biases due to fewer parameters. While some improvements are noted for LLaVA-7B, the highest model scale tested is limited to 7B. There is a valid concern that for much larger or stronger models with greater capacity for long-range contextual understanding and more robust intrinsic cross-modal alignment mechanisms, the benefits of AttAnchor might diminish. The paper does not provide experiments on more advanced models (e.g., Qwen2.5 VL, InternVL 2.5 or LLaVA-OneVision), leaving the extent of AttAnchor's utility in the context of state-of-the-art VLMs uncertain.

**Questions:**

1.  **Robustness to Noisy/Irrelevant Text Content:** Given that AttAnchor relies on cosine similarity to insert text tokens near relevant visual patches, how robust is the method when the input text prompt contains irrelevant, verbose, or even misleading information? Have the authors conducted any experiments or analyses to assess AttAnchor's performance under such "low-quality" text conditions, particularly concerning how cosine similarity might introduce spurious alignments?
2.  **Adaptive Thresholding for $T_{align}$:** The experiments highlight the sensitivity of $T_{align}$. While the paper discusses empirical selection, can the authors propose a more adaptive or dynamic mechanism for determining $T_{align}$ that is less dependent on manual tuning for different tasks and datasets? For example, could an attention-based mechanism or a learned component predict an optimal threshold or dynamically adjust insertion based on context?
3.  **More insights on AttAnchor (Image-into-Text):**  Figure 5 demonstrates that Text-into-Image and Image-into-Text achieve nearly identical training loss; however, their performance during testing diverges significantly, with Image-into-Text often underperforming. Could the authors analyze this discrepancy further?
4.  **Quantitative Analysis of "Semantic Signposts":** Beyond qualitative visualizations in Appendix E, can the authors provide a more analysis or metric to demonstrate how these signposts specifically improve cross-modal grounding and attention scores at deeper layers? For instance, analysis of attention weights distribution or information flow between modalities.

---

### Official Review · Reviewer_v6YF · 2025-11-11

**Soundness:** 2
**Presentation:** 2
**Contribution:** 2
**Rating:** 2
**Confidence:** 4

**Summary:**

This paper proposes AttAnchor, a parameter-free token insertion strategy for VLMs. The method copies each text token and inserts it near the most similar visual token based on cosine similarity, attempting to improve cross-modal attention locality and reduce hallucination.

**Strengths:**

The idea is intuitively appealing and easy to implement.

**Weaknesses:**

- **Lack of Novelty.**

Although the idea is intuitively appealing and easy to implement—and I acknowledge this as a strength—the contribution is marginal and lacks novelty. The core concept of cross-modal token regrouping / routing based on similarity has already been explored in prior literature, especially CrossGET (ICML 2024, “Cross-Guided Ensemble of Tokens for Accelerating Vision-Language Transformers”), which performs cross-guided token selection and grouping to improve multimodal attention.
The paper repeatedly claims novelty such as “first work to cluster cross-modal tokens into shared groups,” which is factually incorrect given existing work. Because of this, the contribution is not strong enough for ICLR, where novelty and positioning in literature are crucial.

- The **technical depth is very limited**.

The method reduces to: compute cosine similarity, then duplicate text token, and insert into token sequence. There is no theoretical grounding or analysis explaining why duplication should improve alignment beyond anecdotal intuition. The mechanism can easily introduce harmful associations if the similarity estimation is noisy. The paper does not discuss failure cases or robustness to negation, ambiguous reference (“it”, “they”), or long text prompts. This level of algorithmic simplicity would be acceptable only if supported by strong and comprehensive experiments — which is unfortunately not the case here.

- **The experimental evaluation is insufficient and selective**.

First, the paper does not compare against CrossGET-like related works, which should be the primary baseline, since both methods manipulate token ordering/grouping to improve cross-modal alignment. The authors instead compare to standard VLMs but ignore the most relevant prior work.

Second, only a limited set of benchmarks is evaluated, and many important task families are missing, e.g., cross-modal retrieval, grounding, or captioning. Third, although the paper claims negligible inference overhead (~0.1%), sequence length is increased due to token duplication, and the paper does not evaluate efficiency under long prompts or multi-image input, where the overhead could be far from negligible.

The ablation study is very shallow; thresholding is tested, but no sensitivity analysis or scaling behavior is shown.

**Questions:**

1. Explicitly compare to CrossGET (ICML 2024) and token routing methods.

2. Provide theoretical / empirical justification for why duplicating text tokens improves alignment.

3. Include experiments on more complex senarios:

Long context,
Multi-image VQA,
Failure modes when similarity signals are noisy.

---

### Note · Program_Chairs · 2026-01-17
**Submission Desk Rejected by Program Chairs**

The following references in this submission do not refer to real documents and/or have major errors in bibliographic information:

     Shengshen Wang, Bowen Dai, et al. Tinyllava: Lightweight language and vision assistant. arXiv preprint arXiv:2311.15102, 2023b.